# External Hoof Measurements of Untrimmed and Unshod Mules in Northern Thailand

**DOI:** 10.3390/ani14081197

**Published:** 2024-04-16

**Authors:** Thawijit Phannithi, Aree Laikul, Watcharapol Pathomsakulwong, Porrakote Rungsri, Tawanhathai Apichaimongkonkun, Krisana Watchrarat, Worakij Cherdchutham

**Affiliations:** 1Veterinary Clinical Studies Program, Faculty of Veterinary Medicine, Kasetsart University, Kamphaeng Saen, Nakorn Pathom 73140, Thailand; thawijit.p@ku.th; 2Department of Large Animal and Wildlife Clinical Sciences, Faculty of Veterinary Medicine, Kasetsart University, Kamphaeng Saen, Nakorn Pathom 73140, Thailand; areelaikul@hotmail.com; 3Equine Clinic, Kasetsart University Veterinary Teaching Hospital, Kasetsart University, Kamphaeng Sean, Nakhon Pathom 73140, Thailand; watgolf2000@gmail.com (W.P.); bellkungvet71@gmail.com (T.A.); 4Equine Clinic, Department of Companion Animal and Wildlife Clinic, Faculty of Veterinary Medicine, Chiang Mai University, Chiang Mai 50100, Thailand; porrakote.rungsri@cmu.ac.th; 5Veterinarian of Royal Stable Unit, The Royal Thai Army, Bangkok 10330, Thailand; kritkrisana@gmail.com

**Keywords:** mule, hoof conformation, forelimbs, hoof balanced, *Equus caballus* × *Equus asinus*, hoof measurement

## Abstract

**Simple Summary:**

The mule has been the partner of humans for millennia, with extensive use in pack saddle and military work. Mule hoof conformation plays a crucial role in the animal’s health, performance, and welfare. Despite the long working history of mules, there has been limited published scientific research on the untrimmed and unshod mule hoof, especially considering its normal shape and sound condition. This study is the first to report on the external hoof conformation of the forelimbs of mules in northern Thailand, accompanied by an assessment of body condition scores. The mules presented homogenous hoof sizes and a high level of symmetry, with small and oval-shaped hooves featuring an upright dorsal wall angle that protrudes more palmar, influenced by the base of the frog. The body condition scores were associated with some external hoof conformation parameters. The study identified and highlighted essential parameters and features of the forelimb hooves in the mule, contributing to a better understanding of this essential aspect of the mule’s external characteristics, which should be useful in hoof management and welfare.

**Abstract:**

External hoof characteristics, balance, and conformation have been extensively studied in horses; however, mules remain understudied in these aspects. This study evaluated the size, shape, and symmetry of untrimmed and unshod forelimb hooves, compared the symmetry between forelimb hooves and stratified external forelimb hoof measurements based on the body condition score of mules raised in the foothill plains of northern Thailand. The forelimb hooves of 38 mules were photographed and 33 parameters, including angular and linear measurements, were analyzed. A multivariate analysis was used to explore the influence of sex, age, and body condition scores (BCS) on angular, linear, and area parameters. Additionally, one-way ANOVA followed by Tukey’s test was used to compare these parameters across different BCS groups. Despite the absence of shoeing and trimming, these mules exhibited optimal left–right forelimb hoof symmetry, with no significant (*p* < 0.05) differences in: outer wall length and inner wall length (OWL–IWL: Left 0.11 ± 0.66 cm; Right −0.12 ± 0.43 cm); sole length and sole width (SL^S^–SW: Left 1.65 ± 0.76 cm; Right 1.46 ± 0.89 cm); dorsal hoof wall length and heel length (DHWL–HL: Left 4.00 ± 0.80 cm; Right 3.81 ± 0.72 cm); and frog length and frog width (FL–FW: Left 3.88 ± 1.13 cm; Right 3.82 ± 0.18 cm). However, significant (*p* < 0.05) differences were observed within each body condition score group for forelimb hoof measurements for DHWL, IWL, heel separation (HS), heel bulb distance (HBD), SW, FW, and FL, while sex and age had no significant differences across the study variables. These findings provide valuable insights into mule welfare and management, contributing to understanding of the interplay between overall health and hoof conformation in the study area.

## 1. Introduction

The mule (*Equus asinus caballus*) is a hybrid offspring from a horse (*Equus caballus*) and a donkey (*Equus asinus*) [1,2,3]. A recent report revealed that the population of mules had rapidly grown as a working animal in some countries such as France, Ethiopia, Peru, and Pakistan [4]. From the literature, mules have been used as pack [5,6] or draught [6,7] animals in hilly terrain or on trails in mountain regions [8], as well as in competitive sporting or gaited competitions [6,9].

The hoof is an essential organ for effective locomotion of equids since it must endure the effects of physical and environmental stresses to safeguard the underlying tissue of the foot from injury and to transfer the force of locomotion to and from the limb skeleton [10]. Therefore, hoof-related defects are an important issue among equine populations and can cause welfare and practical husbandry problems [10]. The hoof conformation in equids is clearly different. For example, the mule inherits its hoof conformation from the donkey, such as the mule’s hoof having an almost upright inclination of hoof wall at the toe, quarter, and heel, while its hoof is smaller, longer, and narrower than for the horse [7]. In addition, hoof conformation varies as a result of changes in the environment, gait, diet, and due to the large amount of management that the hooves may be subjected to [11]. Conversely, hoof conformation prescribes how the foot interacts with the ground and absolutely influences the magnitude and direction of the force entering the limb [12]. Therefore, the genus *Equus* has anatomophysiological compensatory adaptations, such as foot balance [13,14]. Hoof conformation has been studied extensively for horses [11,15,16,17,18,19] and donkeys [11,20], with fewer studies for mules [11]. The shortness or narrowness of hoof characteristics from the forelimb of general purposed horses was related to an increase in movement asymmetry of the limb [17], and the asymmetrical foot shape in horses was proposed to present in foot-related lameness [15]. Furthermore, the hoof conformation of Mongolian horses showed the natural interaction between the hoof and the environment [16]. Souza et al. [11] found that the hooves of horses, mule and donkeys have specific patterns of geometric balance. Thus, the majority of scientific studies have focused on horses, with mules frequently being covered based on extrapolation. The assessment of hoof conformation of mule is one of the keys for maintaining hoof health or in relation to trimming and its treatment. The hoof conformation study from mule could also provide a unique model of the mule managing in various environments, such as conditions with no hoof care or less intensive husbandry, and could demonstrate how a hoof might naturally interact with the environment.

Generally, hoof conformation abnormalities have been described in terms of deviation of the height, angle, or orientation of hoof measurements from defined values, although this approach may fail to consider variation between breeds [21]. In addition, improper hoof balance could be related to the normal or abnormal forces affecting growth, the hoof tissues, distortion, and wear of the hoof [22]. Mediolateral imbalances of the hoof are commonly recognized as a related factor for poor performance and the development of lameness in horses [23,24,25]. Furthermore, chronic hoof misshaping and associated pathological conditions are a critical problem in working horses and draught donkeys because of the hoof health recovery time involved [26,27]. The lack of knowledge regarding the influence of hoof conditions or shape variations on the working capability of equids remains a concern.

In general, sport horses and working donkeys are subjected to a regular hoof trimming program enabling them to perform activities properly. However, the mule has inherited hoof shape and configuration from the donkey [7], which includes small and boxy hooves—a cross between those of horses and the thick sole and tough wall of donkeys [28]. This unique combination of traits equips the mule with exceptional adaptability and functionality. It enables the mule to work effectively in various environments, including urban areas [29] and on steep or otherwise inaccessible terrain [8]. The hoof of a mule can also be trimmed and shod for specific work such as in agriculture, transport, and army service [7]. However, the mule’s hoof is rather upright [7], which may result in different trimming from that applied to a horse. Notably, self-trimming, including cracking, tearing, and increased wear of the hoof under natural herd and environmental conditions, has been suggested as a process for adjusting the hoof conformation of semi-feral ponies, resulting in the front hooves wearing more quickly than the hind hooves [30]. Therefore, the assessment of hoof conformation of mules raised on a specific ground substrate could be one of the keys for maintaining hoof conformation, as well as hoof health and treatment.

The most recent study in hoof disorders in northern Thailand, in 2023, reported on 355 mules [31]. Most of these mules were being used as pack animals for transportation, companionship, or military work. To date, no studies have quantified the hoof conformation from untrimmed and unshod mules in Thailand. Therefore, this study of mules raised in the foothill plains of northern Thailand aimed: (1) to quantitatively assess their external hoof measurements; (2) to compare hoof measurements between their forelimb hooves; and (3) to stratify their external hoof measurements based on body condition scores.

## 2. Materials and Methods

### 2.1. Research Ethics

The study was approved by the Kasetsart University Institutional Animal Care and Use Ethics Committee, Bangkok, Thailand (ID: ACKU65-VET-034).

### 2.2. Study Design and Sample Population

The research design used an observational cross-sectional study of mules in the foothill plains of northern Thailand. The mules were selected through convenience sampling, consisting of 38 healthy mules with no hoof lesions observable by eye and all animals being free from lameness. Lameness was assessed based on a modified five-point scale used for working donkeys [32]. Data on the forelimb hooves from 38 mules were obtained from the Veterinary Remount Department of the Royal Thai Army, Chiang Mai, Thailand that had been bred from crosses predominantly among asses from Thoroughbred mares with male Australian Donkeys. The care takers for each of the mules completed a brief questionnaire providing details of the mule’s age, sex, use, and husbandry.

The body condition score of each mule was evaluated by two veterinarians. Mule body condition scores (BCSs) were assessed based on an equine scoring system, from 0 (very poor) to 5 (very fat) [33]. The BCSs for this study were then recategorized into 4 groups adapted from Carroll et al. [33], based on the distribution of fat across various body parts and practical considerations. Poor condition categories (score 0–1) were characterized by a narrow neck with slack at the base, easily visible ribs, well defined spinous processes, skintight over ribs, or a sunken appearance on either side of the backbone, angular pelvis, supple skin, well defined pelvic and croup, and a deep cavity under the tail. Moderate condition categories (score 2) were characterized by a narrow but firm neck, ribs just visible, a well-covered backbone, felt spinous processes, flat rump on either side of the backbone, well defined croup, and a slight cavity under the tail. Good condition categories (score 3) were characterized by no crest and a firm neck, covered and easily felt ribs, no gutter along the back, covered spinous processes that can be felt, pelvis covered by fat and round with no gutter and easily felt. Obese condition categories (score 4–5) were characterized by a crest, wide and firm neck or folds of fat, well-covered ribs needing firm pressure to feel or cannot be felt, a broad and flat back, a gutter to the root of the tail, and a pelvis covered by soft fat or buried and can be felt with firm pressure or cannot be felt in very fat individuals.

All mule hooves were untrimmed and unshod, with all the mules grazing on naturally growing grass, supplemented fresh cut grass, hay, and occasionally commercial concentrate. Additionally, the mules had never been trimmed and shod in their lifetime. They also accessed drinking water from common water troughs located in the grassy foothill plains, close to the semi-feral habitat [30]. On average, the mules worked three times a week in draught, engaging in tasks such as military equipment packing and carting palanquins. The surface of the mule’s work area was flat, consisting of firm tufted ground covered with a mixture of tiny stones and gravel.

### 2.3. Data Collection

#### 2.3.1. Photographic Method

Digital images were obtained using an iPhone 13 Pro mobile phone camera. Four digital photographs were taken of each forefoot. Prior to capturing the photographs, the mules underwent sedation administered with Acepromazine (Combistress^®^; KELA N.V.; Hoogstraten, Belgium) at a dosage of 0.02–0.04 mg/kg and Xylazine HCl 100 mg/mL (AnaSed^®^ Injection; Akorn, Inc.; Lake Forest, IL, USA) at a dosage of 0.5–1.1 mg/kg body weight through slow intravenous injection to allow for safe handling and accurate positioning of measuring equipment. Lateral, dorsopalmar, and palmarodorsal digital photographs were obtained with the mule standing with both forefeet on the Metron calibration block (https://eponashoe.mybigcommerce.com/metron-block/ accessed on 12 May 2022) and using four calibration points on the block to improve the accuracy of measurement. The hoof position was centered on the block’s centerline. The palmarodorsal view utilized an auto-scaler to automatically set the scale for the image. The auto-scaler was positioned in the plane of the hoof, perpendicular to the camera lens. A white background board that also included a label containing each animal’s identification, including whether the left or right hoof and the date, was placed behind and to the side of the hoof to provide suitable contrast in the image. The center of the camera lens was arranged to focus on the mid-point between the calibration holes on the Metron block for the lateral and dorsopalmar views. Photographic accuracy was enhanced by fixing the iPhone digital camera on an EponaCAM (https://eponashoe.mybigcommerce.com/eponacam/ accessed on 12 May 2022) cradle positioned at least one meter from the Metron block, with zooming according to the guidelines provided by the manufacturer of the Metron block (www.metron-imaging.com accessed on 11 June 2022). Then, the hoof was positioned manually with the camera being perpendicular to the sole, to record the image of the sole. A finger scale ring was positioned next to the solar axes in the plane of the hoof sole. The digital mobile camera was positioned about one meter away and zooming was used to obtain an accurate photographic record.

#### 2.3.2. Image Analysis

The hooves were photographed from the lateral, dorsopalmar, palmarodorsal, and solar views. The digital images were subsequently saved as JEPG files for later analysis using a digital image processing software program called Metron-PX or Metron-Hoof-Pro^®^ Version 8.3.124 with Intellect Module programmed Intellect Engine (IE) version 4.2.1 (copyright 2020 EponaTech; Paso Robles, CA, USA) to objectively assess hoof measurements, based on linear and angular measurements, as shown in Figure 1, along with all the hoof parameter ratios, as shown in Table 1, except for the heel collapse index (HCI). All the digital images were viewed and measured by the same veterinary analyst.

The 35 variables studied consisted of 22 measurements extracted from the digital images, including seven ratio measurements and another five variables created by transformation of measurements taken from the lateral, dorsopalmar, and solar views, to account for individual differences in hoof size and to enable direct comparisons of hoof shape, independent of size effects. The measurements taken are listed in Table 1 and shown in Figure 1.

The last measurement taken (a sole area variable) was based on a grid to calculate the hoof sole area by overlaying a grid on the figure. The sole area was counted and calculated by two of the authors, who were veterinarians.

#### 2.3.3. Accuracy

All digital photographs were taken and measured by one veterinarian. Each view of each hoof was obtained with the camera at a focal distance of approximately 1 m, and zooming was applied in accordance with the guidelines provided by the manufacturer of Metron (www.metron-imaging.com accessed on 11 June 2022) to obtain more accurate photographic data. The camera was set up at the same height as the top of the block and fixed on the EponaCAM cradle. A scale marker was included for palmarodorsal and solar views.

The manufacturer of the Metron software claims that the software is able to make precise measurements of predefined values for the various geometric aspects of the hoof. Another study has reported that using the Metron-PX software is practical and produces measurements with excellent precision and accuracy [36], with mean intra-operator and inter-operator agreement indices exceeding 0.90 for precision and a mean agreement index of at least 0.89 for accuracy.

### 2.4. Statistical Analysis

All analyses were performed using the R Statistical Software (v4.1.2; R Core Team 2021). The Shapiro–Wilk test was used to check the conformation parameters for normality. All data were presented as a mean and standard deviation (SD). The median and interquartile range (IQR) were reported for nonnormally distributed data. The 95% confidence interval (*CI*) was calculated for all parameters. The *t*-test and Wilcoxon signed rank test were used to compare the parametric and non-parametric data between the left and right forelimbs, respectively. A multivariate analysis was conducted using a multiple linear regression model to explore how sex, age, and BCSs collectively influence the hoof parameters (angular, linear, and area parameters). The analysis employed a multiple linear regression model with the hoof parameters as the dependent variable and age, sex, and BCSs as independent variables.

The equation in matrix notation for a multiple regression model was:*Y* = *β*_0_ + *β*_1_*X_i_* + *β*_2_*X_j_* + *β*_3_*X_k_* + *ϵ*
where *Y* is the dependent variables including angular, linear and area parameters. *X_i_* is the age, *X_j_* is sex, and *X_k_* is BCSs. *β*_0_ is the intercept (the value of *Y* when all independent variables (*X_i_*, *X_j_*, *X_k_* are zero). *β*_1_ is coefficient measures the effect of age on *Y*. *β*_2_ is coefficient represents the effect of sex on *Y*. *β*_3_ is coefficient quantifies the impact of BCSs on *Y*, and *ϵ* is the error term capturing the unexplained variability in *Y* not accounted for by the independent variables. One-way ANOVA followed by Tukey’s test to compare angular, linear and area parameters across different BCSs groups (poor, moderate, good, and obese). Significant differences were considered as having a *p*-value less than 0.05.

Within hoof asymmetry was calculated as the differences between: the dorsal hoof wall length (DHWL) and the heel length (HL); the dorsal hoof wall angle (DHWA) and the heel angle (HA); the outer wall length (OWL) and the inner wall length (IWL), the sole length (SLS) and sole width (SW); and the frog length (FL) and frog width (FW).

## 3. Results

This study investigated hoof conformation in a population of draught mules from the foothill plains of northern Thailand. The study consisted of 38 mules, with an age range 4–15 years, with a mean age of 10.45 ± 2.88 years. Specifically, 21 animals were females and 17 were males. The mule BCSs were categorized into four groups: poor condition categories (*n* = 11; 28.95%), moderate condition categories (*n* = 19; 50.00%), good condition categories (*n* = 8; 21.05%), and obese condition categories (*n* = 0; 0.00%). The multivariate analysis indicated that neither age nor sex significantly influenced the studied variables related to external hoof conformation in draught mules. Additionally, the multiple regression analysis showed very low R-squared values for sex or age with angular and linear parameters, suggesting limited explanatory power of these predictors in the model. Overall, these findings strongly suggest that age and sex did not play a significant role in determining external hoof conformation among mules in this region. Given the importance of body condition scores (BCSs), a one-way ANOVA followed by Tukey’s test was conducted to compare angular, linear, and area parameters across different BCS groups (poor, moderate, and good). No preexisting or existing health problems were reported in the mules. Both front hooves were examined in all mules, with no reports from owners regarding routine hoof care or trimming.

### 3.1. Conformational Measurement

The mean, standard deviation, and 95% confidence intervals of each of the variables measured on the forelimb hooves across all mules are shown in Table 2, Table 3 and Table 4.

There were no significant differences between the left and right forefoot external hoof measurement parameters in the mules (*p* < 0.05). Notably, the DHWL tended to differ between the left and right forelimbs (*p* = 0.072). The range and standard deviations for certain dimensions were high; for example, the supporting length varied from 7.89 to 12.72 in the left forelimb and from 8.15 to 12.86 in the right forelimb.

### 3.2. Forelimb Hoof Measurement and Body Condition Scores

The data from all the variables measured for the left and right forelimbs, stratified by body condition scores, are described in Table 5. There were no significant differences between and within each group of BCSs for the forelimbs except in DHWL, IWL, HS, HBD, SW, FW, and FL. The lengths of DHWL and FL were significantly different between groups, with the good and moderate BCSs being higher than for the poor BCSs. The length of IWL was higher in good than in poor BCSs. The lengths of HS, HBD, and FW were higher in poor BCSs. The length of SW was significantly different between groups with poor and moderate BCSs being higher than for good BCSs.

## 4. Discussion

This study explores the hoof morphology of untrimmed and unshod draught mules in the foothill plains in northern Thailand, assesses symmetry between the left and right forelimbs, and stratified morphological data with body condition scores. Despite being a crossbreed of horses and donkeys, mules inherit their hoof conformation and structure predominantly from the donkey, exhibiting characteristics that are smaller, longer, and narrower compared to those of a horse [7]. The hoof of this hybrid achieves an ideal equilibrium by combining the robustness and hardness of the hoof wall with exceptional elasticity, perfectly matching the demands of its natural habitat [7]. Furthermore, the mule has a distinctive upright dorsal hoof wall and broken forward hoof pastern axis [7,37,38].

### 4.1. Conformation

The mules in this study had an average front hoof DHWA of 55.83 degrees (*CI* = 55.09, 56.57), closely resembling the findings of another study by Souza et al. [11] in mules from crosses predominantly among asses from the Pêga breed with Thoroughbred mares or Mangalarga Marchador breeds [11]. The average values for DHWA in Racing Thoroughbreds, Kaimanawa Feral Horses, and Australian Feral Horses are 48.1 degrees [39], 53 degrees [35], and 52.8 degrees [40], respectively. These data suggest that mules tend to have a more elevated hoof angle compared to horses [7]. The inherent influence of the hoof angle on trajectory and landing is noteworthy. An increase in hoof angle is associated with a tendency for the shortening of the cranial stage [11].

In contrast, the average HA of the front hoof observed in this study was 39.95 degrees (*CI* = 34.64, 37.26), lower than the values reported for mules in the study by Souza et al. [11], where the average angle was 44.98 degrees. The HH of the draft mules in this region had an average of 2.36 cm (*CI* = 2.26, 2.45) for the lateral view, 2.01 cm (*CI* = 1.92, 2.09) for LHH, and 1.99 cm (*CI* = 1.90, 2.08) for MHH. These values were lower than the LHH and MHH values of 4.75 and 4.66, respectively, reported in the study by Souza et al. [11].

The angle of the heel should match or closely approximate the angle of the dorsal hoof wall, as suggested by Kauffman and Cline [41], with a low heel angle being indicative of a low heel and often under run heels. Maintaining an average heel height that is neither too low nor too high is crucial, as excessively low heels may signal weakness in the internal supporting tissue [42]. The heels must be the right height to keep the coffin bone properly orientated with the ground [41]. Heel height is also a contributing factor that can lead to a broken forward hoof-pastern axis if the heels are too high, or a broken back hoof-pastern axis if the heels are too low [41].

The differences identified between the present study and the research conducted by Souza et al. [11] were attributed to variations in countries and breeds.

### 4.2. Symmetries in Left and Right Forelimbs

There were no significant differences in the hoof anatomy between the left and right feet in the studied population. Despite no hoof care being provided to the mules in this study, there was a high degree of left–right symmetry observed, similar to the findings by Casanova and Oosterlinck [43] in Catalan Pyrenean horses. However, DHWL tended to be different between the left and right forelimbs. A pair of feet should exhibit considerable similarity, although subtle differences are always present, particularly in the front pair [41]. The hoof wall is a dynamic structure that grows continuously, enabling it to deform in response to applied stresses [44]. Despite the average growth rate of the mule’s hoof wall being approximately 8 mm per month [7], the differences found suggest that a significant portion of this population may exhibit a dominant side or be influenced by other factors, such as changes in hoof angle, that can impact the length of the toe and movement of the center of pressure [17,25,43].

### 4.3. Hoof Symmetry

Asymmetry within the feet was quantified and is illustrated through ratios and differences, as detailed in Table 3 and Table 4.

Mediolateral balance can be observed through the SW_Med_:SW ratio. In this study, the forelimb of the mule had an average of 50.13% or almost equal to 50% of the sole width. Ideally, the medial and lateral width should be balanced, with a 50:50 distribution. Craig [42] observed that in most cases, the lateral side of the hoof tends to be wider, influenced by the shape of the pedal bone. Consequently, some minor asymmetries in the soles are considered quite normal. This can also be observed through the IWH:OWH ratio and OWL–IWL difference, where the average values of these parameters on the forelimbs were 100.44% and 0.01 cm, respectively. This suggested that the length of both the inner and outer walls should remain equal for the hoof to be well-balanced. The pursuit of lateromedial balance is a common practice among farriers. This approach emphasizes that the length of the medial and lateral walls should be equal for the hoof to achieve optimal balance. Furthermore, mediolateral imbalances in the hoof are commonly recognized as contributing factors to poor performance and the development of lameness in horses [23,45,46].

Dorsopalmar balance, assessed through the TL:PL ratio, averaged 44.05% in the forelimbs. An ideal horse foot, according to Kauffmann and Cline [41], positions about two-thirds of its mass behind the true apex of the frog, with 50% of its mass aligned with the widest part of the foot. This location is linked to the center of articulation of the coffin joint in the horse’s hoof. However, Craig [42] suggests using radiographs for more accessible landmark identification. It would be of interest to conduct future research involving radiographs to determine the location of this point of balance in mules. The SL^S^–SW measured values of the forelimbs showed that SL^S^ values were longer than for SW, suggesting that the mule hoof is oval shaped.

In the current study, there was a lack of parallelism in the guidelines, similar to the findings in the studies by Souza et al. [11] and Khan et al. [32], deviating from the unanimous agreement across the literature that the ideal alignment, as viewed laterally, involves the parallel positioning of the hoof and heel [46,47,48]. Craig [49] found that most horses have good support length, with a hoof angle only slightly larger than the heel angle; however, the difference should not exceed 10 degrees. Another useful index for defining the hoof shape is the HCI [12], as indicated by Eliashar et al. [12], with a ratio of 1.00 defines heel angle and dorsal hoof wall angle that remain parallel. In the present study, the mule presented an average heel collapse of 0.65 (64.52%), indicating that the line segments of the dorsal hoof wall and the heel extend distally, resulting in a reduction in their horizontal length. This reduction is defined as a divergence of the hoof wall with a heel collapse index of <1.00, as reported by Clayton et al. [50], McClinchey et al. [51], Eliashar et al. [12], and Dyson et al. [19].

A defective hoof capsule, being more susceptible to infection from the environment and vulnerable to damage due to improper force transmission during locomotion [10], can lead to both welfare and practical husbandry problems [10]. However, notably, all mules in the present study were utilized as draught animals, engaging in activities distinct from those of sport horses. Consequently, the primary causes of foot lameness in mules may likely differ from those affecting sport horses.

### 4.4. Comparison of the External Hoof Characteristics Stratified by Body Condition Scores

The structure of the mule’s hoof is optimally adapted to efficiently bear the weight and pressures in proportion to the animal’s size, the characteristics of the terrain it traverses, and the specific nature of its gaits [7]. Consequently, it is possible to observe changes in the shape of the hoof in various areas, each exhibiting distinct modalities and occurring at different times in response to weight. The mule appears to predominantly inherit its hoof conformation and structure from the donkey, as previously discussed. In line with this, Thiemann et al. [38] observed that donkeys bear weight in a five-point pattern, distributing pressure at each heel and the toe, with additional weight-bearing occurring on the sole at the toe. Turner [44] further emphasized the influence of horse weight on foot dynamics, proposing that body condition scores can provide a useful estimate in this regard. BCSs seem to act as triggers for changes in certain external hoof parameters.

The discovery of a significant effect of BCSs on DHWL is noteworthy. The average DHWL for forelimbs is 7.26 cm, a value in close alignment with the study on equids by Souza et al. [11], where the value is the same in donkeys but slightly shorter in mules. The DHWL appears to be higher in mules with good and moderate BCSs compared to those with moderate and poor BCSs. This observation aligns with Turner’s study, suggesting that increased weight is related to higher DHWL. Another factor is the length of the dorsal hoof wall, with studies showing that the forces applied on the foot were correlated to the changes in the ratio of heel-to-dorsal hoof wall heights [12,52]. Turner [44] recommends that the length of the heel should typically constitute about one-third of the length of the dorsal hoof wall. Building on this recommendation, our study findings reveal that mules exhibit a HL exceeding one-third of the dorsal hoof wall length (HL:DHWL 46.35%).

Ideally, the symmetry of the hoof should have the frog dividing the solar surface into equal medial and lateral parts [22]. Investigating the interplay between weight-bearing and BCSs revealed a significant role played by both FL and FW. Mules with good and moderate BCSs exhibit a greater FL compared to those with poor BCSs, while the FW in mules with poor BCSs appears higher than those with good BCSs. The mule averages 7.81 cm for FL and 3.95 cm for FW, with a 3.85 cm difference between them. FL aligns with Souza et al. [11], while FW is slightly shorter in our findings. According to Turner [44], in horses, the ideal FW should be two-thirds of its length, while FL should be two-thirds of the SL^S^. However, in our study, mule frogs appear narrower than these proportions suggest, and the FL seems to be higher than typically observed in horses. In their study, Kauffmann and Cline [41] observed that in climates with adequate moisture, a healthy frog tends to be broad and plump, with a rubbery texture. In contrast, in environments characterized by low moisture and rugged terrain, the frog may undergo changes, such as increased dryness, hardness, narrowing, and a more vaulted shape, as an adaptation. In the present study, all mules grazed on the foothill plains, suggesting that this type of substrate influenced the shape of the frog. This adaptation is crucial, considering the frog’s essential role in maintaining the anterior/posterior balance of the hoof. It serves a vital function in hoof biomechanics, influencing the alignment of the distal phalanx and the pastern. The importance of the frog becomes particularly noticeable during weight-bearing, as highlighted in the study by Ovnicek et al. [53].

The SW results in the present study align with Souza et al. [11] findings for mules, while the average SL^S^, though shorter, corresponds to values observed in donkeys. Additionally, the SL:SW ratio of the hoof is higher than in horses, based on Souza et al. [11], with their elaboration that this difference is due to the measurement being higher in mules and donkeys because the base of the frog protrudes more into the palmar/plantar region. Not just the frog, but also SW is another factor that is related to BCSs. Furthermore, there is a significant difference in SW among BCS groups. Mules with poor and moderate BCSs had wider soles compared to those with moderate and good BCSs. SA was not significantly different among BCSs; however, a study in riding school horses indicated that increased body size has a greater influence on the conformation of the hooves, with hoof base width appears to have larger hooves or increase in greater solar surface area [54]. The loading of hooves has been documented, where excessive weight in a small area can elevate the risk of lameness, [55,56]. In instances of high body condition, a critical value greater than 5.5 kg/cm^2^ in adult horses, as identified by Turner [21], indicates that the hoof size may be too small. Nevertheless, the outcomes for the mule are currently unavailable. In the present study, the length of HS and HBD in the mules with poor BCSs was higher than in those with good BCSs. If the heel bulb distance is excessively large, and the frog exhibits a stretched and thin appearance, these characteristics are indicative of hooves with contracted heels. Furthermore, a substantial heel bulb distance not only signals contracted heels but also serves as an indicator of the positioning of the sole body. This suggests that the sole body is not aligned under the bony column [42].

The mules in the present investigation, located in the foothill plains of northern Thailand, had not undergone any form of foot care, trimming, or shoeing. As a result, their hooves could be deemed to be in a naturally shaped state [57]. In the context of wild horses, it has been observed that natural hoof care contributes to the attainment of an optimal form for proper function and soundness [53]. The mule in the present study had homogenous hoof sizes and a high level of symmetry with a small and oval shape. The high level of symmetry observed in the mules’ hooves may be attributed, at least partially, to continuous and unrestricted movement across a variety of substrates, a diverse diet, and natural selection in harsh environmental conditions. Importantly, the appearance of this foot conformation does not require the trimming like feral horses [43]. Additionally, the 0 to 5 system body condition score [33] was adapted in this study to assess the mules’ physical condition based on the authors’ preliminary observations, which showed that there were no mules classified as having a very poor or very fat body condition. The four categories of body condition scores in the study were used due to its simplicity and practicality in evaluating body fat deposition across various body parts. However, the body condition score in our study is divided into categories, so correlation analysis will not be performed.

The study’s limitations include reliance on convenience sampling and the assessment of mules engaged in specific work, particularly those with healthy hooves and no observed hoof lesions, based on a visual inspection. Lameness evaluation relied on predefined criteria due to the mules’ reluctance to trot. This approach may have introduced errors, particularly due to the absence of lameness sensors for detection and the lack of specific lameness guidelines, especially for mules. Notably, the mules in this study were not trained for trotting, which added complexity to lameness detection in this population. Moreover, the height at withers and circumference of the chest information in our study are limited conditions to collect because mules become excited even in a sedated state and may attempt to kick or become scared of unfamiliar people and equipment, posing risks to both the safety of the animals and humans. Furthermore, heavy sedation of the mules will affect their standing balance while collecting data. Further work should assess mule liveweight correlation with hoof conformation based on the relationship between the animal’s body mass and hoof parameters [20]. Additionally, exploring the potential of radiographic techniques may contribute to a more comprehensive understanding of mule hoof conformation.

## 5. Conclusions

To the author’s knowledge, there is limited information regarding untrimmed and unshod mule hooves in the literature. However, it is noted that mules typically present homogenous hoof size, shape, and a high level of symmetry for forelimb hooves. Therefore, the present study, involving external hoof measurements in a mule population from the foothill plains without hoof care or routine trimming, may provide a specific model representing how a mule hoof might conventionally interact with this unique environment. Additionally, describing the BCSs provides insights into their influence on external hoof morphometric changes. Furthermore, this study provides an essential baseline photographic interpretation of mules, which could be clinically important, providing guidelines for the proper trimming of their hooves in the future and for assessing lameness to improve mule welfare due to the impact on workload, with potential benefits for industries such as agriculture and transportation.

## Figures and Tables

**Figure 1 animals-14-01197-f001:**
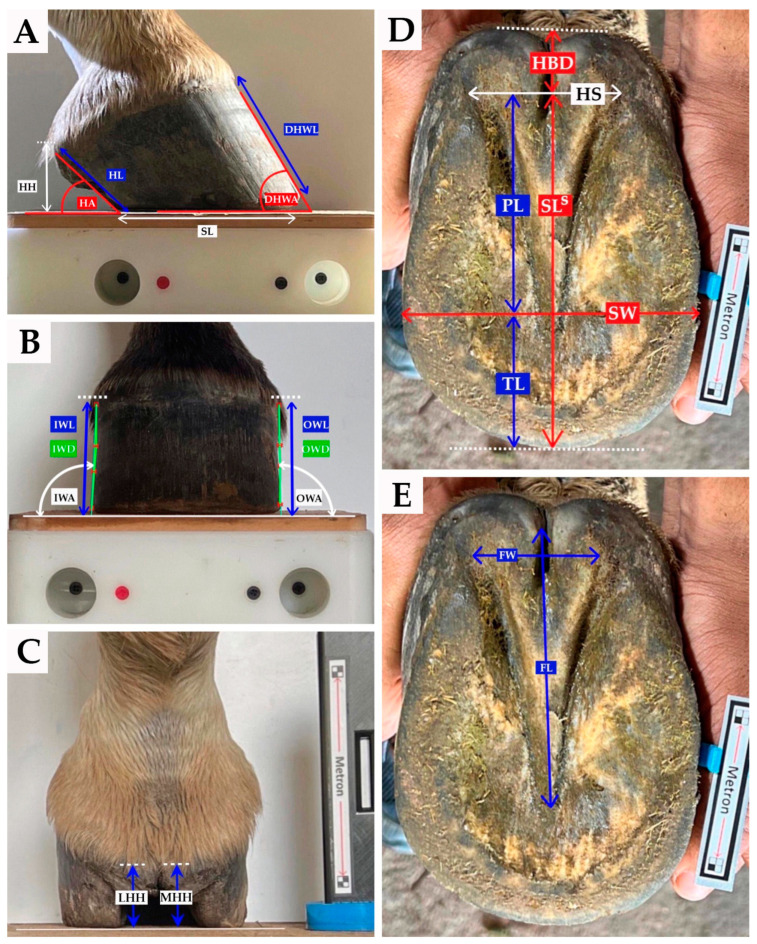
External hoof measurements of fore digit in mule shown in (**A**) lateral view, (**B**) dorsopalmar view, (**C**) palmarodorsal view, (**D**,**E**) solar view. Abbreviations are listed in Table 1.

**Table 1 animals-14-01197-t001:** Abbreviations and definitions of linear and angle measurements acquired and calculated ratios and calculated differences of hoof measurements (Adapted from Cruz et al. [34], Dyson et al. [15] and Hampson et al. [35]).

Variable	Abbreviation	Description of Measurement
Angular parameters		
Dorsal hoof wall angle	DHWA	Angle subtended between the dorsal hoof wall and the ground surface.
Heel angle	HA	Angle subtended between the palmar aspect of the hoof and the ground surface.
Outer wall angle	OWA	Angle subtended between the dorsal lateral angle between the abaxial aspects of the hoof wall, in dorsal view.
Inner wall angle	IWA	Angle subtended between the dorsal medial angle between the abaxial aspects of the hoof wall, in dorsal view.
Linear parameters		
Dorsal hoof wall length	DHWL	Length from the hairline along the dorsal surface of dorsal hoof wall to the tip of the hoof.
Heel height	HH	Vertical distance between the palmar aspect of the lateral heel at the level of the coronary band and the ground surface.
Heel length	HL	Linear length of palmar hoof wall to the distal limit of the palmar hoof wall at the weight bearing boarder.
Support length	SL	Horizontal distance between the dorsal-most point and the palmar-most point of weight-bearing surfaces of the foot, in lateral view.
Heel separation	HS	Horizontal distance between the heel points support that are on the hoof wall (the distance between the heel points).
Heel bulb distance	HBD	Linear distance from the heel to the torus ungulae.
Sole width	SW	Linear distance between the widest point of the sole.
Sole length	SL^S^	Vertical distance between the center of the toe to the heel buttress line (excluding the heel bulb), in solar view.
Toe length	TL	Linear distance of the center of the toe (solar view) to the center of the widest point of the sole.
Palmar length	PL	Linear distance between the center of the widest part of the sole to the center of the heel separation.
Frog width	FW	Linear distance at the widest point of the frog.
Frog length	FL	Linear distance between the tip of the apex cunei (point of frog) and the line connecting the palmar ends of the cuneus ungulae (base of frog).
Outer wall length	OWL	Linear distance from the hairline to the ground, on the abaxial aspects of the lateral hoof wall, in dorsal view.
Inner wall length	IWL	Linear distance from the hairline to the ground, on the abaxial aspects of the medial hoof wall, in dorsal view.
Outer wall deviation	OWD	Imaginary line distance between the same angle all the way from the coronet to the ground, on the abaxial aspects of the lateral hoof wall.
Inner wall deviation	IWD	Imaginary line distance between the same angle all the way from the coronet to the ground, on the abaxial aspects of the lateral hoof wall.
Lateral heel height	LHH	Linear distance from the hairline at the palmar most aspect to the lateral heel bulb to the ground.
Medial heel height	MHH	Linear length from the hairline at the palmar most aspect to the medial heel bulb to the ground.
Transformation of measurement		
Differences		
Dorsal hoof wall to heel angle difference or Degree of heel collapse (DHC)	DHWA-HA or DHC	Angular difference between dorsal hoof wall and heel angles.
Outer wall to inner wall length difference	OWL–IWL	Length difference between outer wall length and inner wall length.
Sole length to sole width difference	SL^S^–SW	Length difference between sole length and sole width.
Dorsal hoof wall to heel length difference	DHWL–HL	Length difference between dorsal hoof wall length and heel length.
Frog length to frog width difference	FL–FW	Length difference between frog length and frog width.
Ratios		
Heel collapse index	HCI	Ratio between heel angle and dorsal hoof wall angle.
Heel/sole width ratio	HS:SW	Ratio between heel separation and sole width.
Heel bulb distance/Support length ratio	HBD:SL^S^	Ratio between heel bulb distance and supporting length from solar view.
Toe/Length ratio	TL:PL	Ratio between linear distance of the center of the toe to the center of the widest part of the sole and linear distance between the center of the widest part of the sole to the center of the heel separate.
Medial sole width/Sole width ratio	SW_Med_:SW	Ratio between linear distance of Pars medialis/widest part of sole width.
Inner/Outer height ratio	IWH:OWH	Ratio between the vertical distance from the dorsal medial coronary band to the ground surface divided by the vertical distance from the dorsal lateral coronary band to the ground surface.
Heel length/Dorsal hoof wall length	HL:DHWL	Ratio between heel length and dorsal hoof wall length.
Area parameter		
Sole area	SA	Total area of the sole area excluding the area of the frog.

**Table 2 animals-14-01197-t002:** Description of external hoof measurement for forelimbs of 38 mules, comparing left forelimb to right forelimb. Abbreviations are listed in Table 1.

	Left Forelimb (*n* = 38)				Right Forelimb (*n* = 38)				*p*-Value
	Mean ± SD	Min	Max	*CI*	Mean ± SD	Min	Max	*CI*	
Measured on Lateral view (see Figure 1A)									
DHWA	55.08 ± 3.42	49.58	64.83	[54.98, 57.18]	55.57 ± 3.13	48.93	63.37	[54.55, 56.57]	0.498
DHWL	7.32 ± 0.09	6.09	8.47	[7.13, 7.50]	7.20 ± 0.09	5.99	8.61	[7.01, 7.39]	0.072
HA	36.25 ± 6.53	22.03	54.18	[34.14, 38.35]	35.65 ± 4.92	26.60	45.86	[34.06, 37.24]	0.835 ^A^
HH	2.40 ± 0.37	1.82	3.56	[2.27, 2.51]	2.32 ± 0.40	1.44	3.19	[2.18, 2.44]	0.354
HL	3.31 ± 0.65	2.22	4.57	[3.11, 3.52]	3.39 ± 0.57	2.33	4.65	[3.21, 3.58]	0.424
SL	10.09 ± 1.07	7.98	12.72	[9.74, 10.44]	10.28 ± 1.19	8.15	12.86	[9.90, 10.65]	0.484
Measured on Dorsopalmar view (see Figure 1B)									
OWL	5.53 ± 0.50	4.36	6.60	[5.36, 5.68]	5.55 ± 0.53	4.30	6.69	[5.37, 5.71]	0.885
IWL	5.64 ± 0.66	3.69	6.90	[5.42, 5.85]	5.45 ± 0.51	4.52	6.64	[5.28, 5.61]	0.176
OWA	94.46 ± 5.56	84.81	109.33	[92.67, 96.25]	93.33 ± 8.95	48.40	105.48	[90.44, 96.21]	0.835 ^A^
IWA	95.08 ± 5.46	83.03	107.80	[93.31, 96.83]	93.88 ± 5.82	85.43	110.57	[92.00, 95.75]	0.365
OWD	0.30 ± 0.20	0.05	0.96	[0.23, 0.36]	0.35 ± 0.30	0.02	1.67	[0.24, 0.44]	0.611 ^A^
IWD	0.37 ± 0.17	0.03	0.75	[0.31, 0.42]	0.45 ± 0.32	0.03	1.38	[0.34, 0.55]	0.611 ^A^
Measured on Palmarodorsal view (see Figure 1C)									
LHH	2.02 ± 0.36	1.40	3.00	[1.90, 2.13]	2.00 ± 0.43	1.10	3.50	[1.86, 2.13]	0.499 ^A^
MHH	1.99 ± 0.38	1.40	2.80	[1.86, 2.10]	2.00 ± 0.41	1.10	2.90	[1.86, 2.13]	0.877
Measured on Solar view (see Figure 1D,E)									
HS	4.56 ± 0.72	2.93	6.38	[4.32, 4.78]	4.62 ± 0.66	3.26	6.11	[4.41, 4.83]	0.599
HBD	1.97 ± 0.65	1.08	4.04	[2.61, 2.99]	2.11 ± 0.60	1.20	3.74	[2.49, 2.94]	0.175 ^A^
SW	9.31 ± 0.69	7.89	10.93	[9.09, 9.53]	9.40 ± 0.70	8.25	11.97	[9.17, 9.63]	0.678 ^A^
SL^S^	10.96 ± 0.88	8.04	12.79	[10.67, 11.24]	10.87 ± 1.01	8.66	13.13	[10.54, 11.19]	0.454 ^A^
FW	3.93 ± 0.10	2.79	5.26	[3.72, 4.13]	3.99 ± 0.11	2.90	5.60	[3.76, 4.20]	0.562
FL	7.82 ± 0.13	6.05	9.19	[7.54, 8.08]	7.81 ± 0.13	6.29	9.87	[7.53, 8.07]	0.938
SA	56.71 ± 9.32	39.79	74.30	[53.74, 59.67]	56.32 ± 7.74	41.79	76.88	[53.85, 58.78]	0.795

Note: For data not normally distributed, median and interquartile range are shown; the Wilcoxon signed-rank test *p*-value was used: Left foot: HA 34.57 ± 7.11; HBD 1.71 ± 1.22; SLS 11.04 ± 1.12; OWA 93.50 ± 6.45; OWD 0.22 ± 0.32; Right foot: HBD 1.99 ± 0.80; SW 9.35 ± 0.71; OWA 94.36 ± 7.24; OWD 0.26 ± 0.29; IWD 0.36 ± 0.36; LHH 1.90 ± 0.40. ^A^ Wilcoxon signed rank test *p*-value. *CI*, confidence interval. All linear measurements are in centimeters, angles are shown in degrees and sole measurements are in square centimeters.

**Table 3 animals-14-01197-t003:** Description of transformation of external hoof measurement taken from lateral, dorsopalmar, and solar views for 38 mules, comparing left forelimb to right forelimb.

	Left	Right	*p*-Value	*CI*	
	Mean (SD; Range)	Mean (SD; Range)		Left	Right
Hoof parameters (%)					
HCI	64.79 (11.87; 39.82–92.19)	64.24 (8.74; 44.78–81.98)	0.952	[61.02, 68.57]	[61.47, 67.03]
HS:SW	48.92 (6.43; 31.57–61.83)	48.93 (5.98; 34.15–61.40)	0.991	[49.63, 50.65]	[49.76, 50.47]
HBD:SL^S^	18.21 (6.93; 9.20–44.10)	19.78 (6.75; 11.46–43.16)	0.220 *	[15.97, 20.44]	[17.60, 21.95]
TL:PL	43.85 (6.34; 27.45–60.96)	44.26 (4.98; 30.36–55.93)	0.762	[41.81, 45.89]	[42.65, 45.86]
SW_Med_:SW	50.14 (1.58; 45.94–53.10)	50.12 (1.11; 47.27–52.18)	0.941	[49.63, 50.65]	[49.76, 50.47]
IWH:OWH	102.23 (10.74; 70.87–125.21)	98.64 (7.64; 76.35–112.77)	0.102	[98.77, 105.69]	[96.17, 101.10]
HL:DHWL	45.43 (8.76; 27.97–58.51)	47.26 (7.85; 31.32–64.17)	0.185	[42.65, 48.22]	[44.77, 49.76]
Differences in hoof parameters					
Angle (°)					
DHC	19.83 (7.08; 4.42–33.29)	19.91 (5.24; 10.08–32.80)	0.952	[17.58, 22.08]	[18.25, 21.58]
Hoof wall height (cm)					
OWL–IWL	0.11 (0.66; −1.81–1.52)	−0.12 (0.43; −1.25–0.68)	0.141	[−0.09, 0.32]	[−0.24, 0.05]
Sole (cm)					
SL^S^–SW	1.65 (0.76; −0.39–3.04)	1.46 (0.89; −1.08–3.19)	0.087	[1.41, 1.89]	[1.18, 1.75]
Length (cm)					
DHWL–HL	4.00 (0.80; 2.78–6.00)	3.81 (0.72; 2.34–5.63)	0.097	[3.75, 4.26]	[3.58, 4.04]
Frog (cm)					
FL–FW	3.88 (1.13; 1.81–5.35)	3.82 (0.18; 1.68–5.31)	0.604 *	[3.52, 4.25]	[3.47, 4.17]

Note: For data not normally distributed, median and interquartile range are shown, the Wilcoxon signed-rank test *p*-value was used: Left foot: HBD:SL 16.43 ± 8.11; HL:DHWL 45.58 ± 17.13; FL–FW 4.19 ± 2.04; Right foot: HBD:SL 17.77 ± 9.05; FL–FW 3.89 ± 1.49. Abbreviations are listed in Table 1. * Wilcoxon signed rank test *p*-value.

**Table 4 animals-14-01197-t004:** Description of external hoof measurements of hooves for forelimbs of 38 mules.

Measurement	Mean ± SD	*CI*
Measured on Lateral view (see Figure 1A)		
DHWA	55.83 ± 3.31	[55.09, 56.57]
DHWL	7.26 ± 0.59	[7.13, 7.39]
HA	35.95 ± 5.83	[34.64, 37.26]
HH	2.36 ± 0.39	[2.26, 2.45]
HL	3.36 ± 0.61	[3.21, 3.49]
SL	10.19 ± 1.13	[9.93, 10.44]
Measured on Dorsopalmar view (see Figure 1B)		
OWA	93.89 ± 7.52	[92.21, 95.59]
OWD	0.32 ± 0.26	[0.26, 0.38]
OWL	5.53 ± 0.52	[5.42, 5.65]
IWA	94.48 ± 5.71	[93.19, 95.76]
IWD	0.41 ± 0.26	[0.35, 0.47]
IWL	5.55 ± 0.60	[5.41, 5.68]
Measured on Palmarodorsal view (see Figure 1C)		
LHH	2.01 ± 0.39	[1.92, 2.09]
MHH	1.99 ± 0.39	[1.90, 2.08]
Measured on Solar view (see Figure 1D,E)		
HS	4.59 ± 0.69	[4.43, 4.75]
HBD	2.77 ± 0.65	[2.62, 2.91]
SW	9.36 ± 0.70	[9.20, 9.52]
SL^S^	10.91 ± 0.95	[10.69, 11.13]
FW	3.95 ± 0.66	[3.81, 4.11]
FL	7.81 ± 0.84	[7.62, 8.00]
SA	56.51 ± 8.51	[54.60, 58.43]
Hoof parameters (%)		
HCI	64.52 ± 10.36	[62.19, 66.85]
HS:SW	48.92 ± 6.25	[47.51, 50.33]
HBD:SL^S^	18.99 ± 6.93	[17.44, 20.55]
TL:PL	44.05 ± 5.74	[42.76, 45.35]
SW_Med_:SW	50.13 ± 1.38	[49.82, 50.44]
IWH:OWH	100.44 ± 9.55	[98.29, 102.58]
HL:DHWL	46.35 ± 8.31	[44.48, 48.22]
Differences in hoof parameters		
DHC (°)	19.87 ± 6.19	[18.48, 21.26]
OWL–IWL (cm)	0.01 ± 0.57	[−0.12, 0.14]
SL^S^–SW (cm)	1.56 ± 0.83	[1.37, 1.74]
DHWL–HL (cm)	3.91 ± 0.76	[3.74, 4.08]
FL–FW (cm)	3.85 ± 1.10	[3.61, 4.10]

Note: For data not normally distributed, median and interquartile range are shown: SL 9.99 ± 1.47; IWD 0.36 ± 0.27; OWD 0.23 ± 0.31; OWA 93.67 ± 6.86; LHH 2.00 ± 0.43; HBD 1.90 ± 0.78; SW 9.29 ± 0.81; SW_Med_:SW 50.11 ± 1.45; HBD:SL^S^ 17.39 ± 8.41; FL–FW 4.02 ± 1.60. Abbreviations are listed in Table 1. *CI*, confidence interval. All linear measurements are in centimeters, angles are shown in degrees and sole measurements are in square centimeters.

**Table 5 animals-14-01197-t005:** Mean and standard deviations of external hoof measurements for forelimbs stratified by body condition scores of 38 mules.

Measurement	Body Condition Score						*p*-Value
	Poor	No.	Moderate	No.	Good	No.	
Lateral view (see Figure 1A)							
DHWA	54.65 ± 3.60	22	56.08 ± 3.34	38	56.84 ± 2.41	16	0.105
DHWL	6.95 ± 0.66 ^b^	22	7.33 ± 0.52 ^a^	38	7.53 ± 0.53 ^a^	16	0.007
HA	36.80 ± 5.70	22	35.86 ± 5.14	38	35.01 ± 7.57	16	0.646
HH	2.24 ± 0.48	22	2.43 ± 0.31	38	2.35 ± 0.41	16	0.211
HL	3.19 ± 0.50	22	3.42 ± 0.54	38	3.42 ± 0.87	16	0.331
SL	10.19 ± 0.96	22	10.06 ± 1.17	38	10.47 ± 1.25	16	0.483
Dorsopalmar view (see Figure 1B)							
OWA	95.38 ± 12.57	22	93.88 ± 4.41	38	91.91 ± 2.57	16	0.380
OWD	0.36 ± 0.21	22	0.28 ± 0.21	38	0.38 ± 0.40	16	0.341
OWL	5.51 ± 0.57	22	5.53 ± 0.48	38	5.59 ± 0.57	16	0.897
IWA	93.90 ± 7.48	22	95.00 ± 4.92	38	94.02 ± 4.84	16	0.729
IWD	0.40 ± 0.29	22	0.40 ± 0.24	38	0.45 ± 0.28	16	0.802
IWL	5.26 ± 0.58 ^b^	22	5.61 ± 0.55 ^ab^	38	5.80 ± 0.61 ^a^	16	0.014
Palmarodorsal view (see Figure 1C)							
LHH	2.05 ± 0.39	22	1.91 ± 0.35	38	2.18 ± 0.46	16	0.06
MHH	2.03 ± 0.41	22	1.92 ± 0.36	38	2.11 ± 0.44	16	0.220
Solar view (see Figure 1D,E)							
HS	4.81 ± 0.84 ^a^	22	4.60 ± 0.62 ^ab^	38	4.16 ± 0.55 ^b^	16	0.019
HBD	2.35 ± 0.75 ^a^	22	1.96 ± 0.56 ^ab^	38	1.84 ± 0.44 ^b^	16	0.023
SW	9.48 ± 0.82 ^ab^	22	9.43 ± 0.71 ^b^	38	8.71 ± 0.32 ^c^	16	0.013
SL^S^	10.48 ± 1.02	22	11.03 ± 0.93	38	11.08 ± 0.74	16	0.082
FW	4.27 ± 0.75 ^a^	22	3.92 ± 0.58 ^ab^	38	3.63 ± 0.53 ^b^	16	0.008
FL	7.10 ± 0.70 ^c^	22	8.04 ± 0.80 ^b^	38	8.26 ± 0.42 ^ab^	16	< 0.001
SA	58.77 ± 8.94	22	56.16 ± 8.60	38	52.11 ± 7.39	16	0.096

Note: ^a,b,c^ Different lowercase superscripts in row indicate significant differences (*p*-value < 0.05). Abbreviations are listed in Table 1. All linear measurements are in centimeters, angles are shown in degrees and sole measurements are in square centimeters.

## Data Availability

The data presented in this study are available on request from the corresponding author. The data are not publicly available due to privacy concerns.

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
