# Peer review of "External Hoof Measurements of Untrimmed and Unshod Mules in Northern Thailand"

_animals, 2024, doi:10.3390/ani14081197_

Round 1

Reviewer 1 Report

Comments and Suggestions for Authors

Line 59: "ideal" is relative to possible function or adaptative, can you be more precise about why you state the mule receives some ideal hoof conformation from the donkey

Line 73: what justifies the "Thus" as if what precedes implies the rest of this sentence

Line 88 what arguments justifies this statement? or you need to be more explicit

Line 119 age range should be in result not material and method

Lines 123-124 although more recently published the grading out of 5 for BCS is less widely used in horses then the out of 9 scale, can you comment on your choice about the out of 5 scale method for this project, probably in the discussion section

In statistical analysis can you address the power calculation of your project to justify the sample size / number of animal included

Line 209: would you rather use animal or individual than "beasts"

Lines 210 to 213 would you reword with poor condition, average condition etc, as "poor animal" "good animal" ... isn't as semantically exact

Line 272, p=0.072 is not statistically significant so you can't use this nomenclature, talk about trend but not significance

Line 296 under run heels? instead of under heel?

Line 369 do you mean pattern instead of pastern?

This is a well written paper with sound rationale and strong narrative

Very novel information useful to increase working equids welfare and husbandry. Adresses correctly limitations and future research/investigation directions

Comments on the Quality of English Language

Minor edits reported in prior section together with other comments

Author Response

Dear Reviewer 1

Thank you very much for your helpful information and suggestions. Now, I have addressed your suggestive and necessary points from your comment. I have already attached the revised version with this box.

Sincerely,

Reviewer 2 Report

Comments and Suggestions for Authors

  I have no complaints about the discussion. However, in your conclusion, I could not understand how knowing the relationship between physical condition scores and the shape of a mule's hooves could be connected to the industry of your country. In other words, I thought it was unclear what specifically the results of this research would be applied to. I would like you to comment on specific ways you can contribute to the industry.

Author Response

Dear Reviewer 2,

Thank you very much for your helpful information and suggestions. Now, I have addressed your suggestive and necessary points from your comment. I have already attached the revised version with this box.

Sincerely,

Reviewer 3 Report

Comments and Suggestions for Authors

animals 2920313

The paper concentrates on the hoof aspect of healthy mules, their size, and connections with body score evaluation. It is an interesting paper, quite well-written, and informative. However, some aspects should be solved/re-written before publishing:

  1. Measurements should be described as detailed as possible in Table 1 (exactly points should be given as much as possible) and marked better on the photos attached – figure 2 (the lines seem too small, not visible); 
  2. The analysis of variance seems only a one-factor analysis. That is not clear from the description. If it is one-way ANOVA? Why multifactorial analysis was not used? Sex and age differences were not studied that way.

Detailed remarks:

statistics should be described in the abstract 

L 59 it is difficult to agree that narrow hoofs can be the ideal hoofs for any working animal – the base of weight per area is smaller, the forces are higher per cm2

L 63 – use “gait” instead of “pace”, as the pace is the name of the special horse gait

L 73 – describe more – citation 16 – based on…

L 77-79 – and the other connection also exists 

L 91 – give a citation for the upright position of the mule hoof

L 94 – please connect your idea with the cited text – is a mule a semi-feral animal?

L 114- that is not clear why you cite the number of animals? Why 30 as you have 38?

L 119 – A description of your animals is not enough – what about their height at withers and circumference of the chest?

L 121 –why did not you check all these factors? Why sex/age differences are not investigated?

L 122 – please exclude here and in any other places the names of people doing these parts or experiments. There is a special part of the paper - overall information at the end of the paper. there is a place here for all author's remarks. L 164/173/182 etc.

L 125 – how long they were not trimmed? In a life?

L 129 – what was the surface of their work?

L 150 – have you checked the precision of your system by using the exact known form and measuring it? That would be interesting.

Table 1 – more details would be useful

Figure 1 – perhaps more figures will be useful with better-marked lines

L 193 – precision should be described in more detailed

L 199 multi factorial? Not clear. Age could be a regression, sex should be a new class.

L 209 beast? Used better – animal.

Table 4 should be a supplement.

 L 362 – correlations were not investigated. You have here the influence of the BCS on the hoof measurements. There was nothing on correlations in your statistical parts.

L 379 – was it the correlation?

Conclusions should concentrate more on your own results – more on measurements should be given in the overall, general way (perhaps mentioned in groups).

Author Response

Dear Reviewer 3,

Thank you very much for your helpful information and suggestions. Now, I have addressed your suggestive and necessary points from your comment. I have already attached the revised version with this box.

Sincerely,

Round 2

Reviewer 3 Report

Comments and Suggestions for Authors

Some corrections were made, however the paper needs further work.

The information given in the Author's response should be included in the text as much as possible for other readers. So, please include the information on the preliminary statistical analysis in the paper as well as BSC scoring. The abstract was not corrected - information on the statistical method used should be included.  

The information on the optimum hoof was corrected in L59, but not L94.

Please divide your text between tables - tables should not follow each other without the text. 

The conclusions seem too long. Too many results in conclusions. They are not real conclusions. Do not repeat results with shortcuts of variables. Your conclusions should be able to be read without the main text and more general in meaning.  

Author Response

Dear Reviewer 3,

We have carefully considered the suggestions provided and have implemented them in the second revision. I have already attached the revised version with this box. Thank you very much for your thoughtful feedback.

Kind regards,

Worakij

Round 3

Reviewer 3 Report

Comments and Suggestions for Authors

The paper still is interesting and worth publishing, however not corrected. The paper has to be clear and repeatable – the Authors do not correct the paper in the way that allows for it.

The BSC is not described in detail enough, it is not clear what is "good condition" and what is "poor" according to these sentences:

“The body condition score of each mule was evaluated by two veterinarians. Mule 131 body condition scores (BCSs) were assessed based on an equine scoring system, from 0 132 (poor) to 5 (obese) [33]. The BCSs for this study were categorized into 4 groups, where 133 score of 0 and 1 represent poor condition categories, a score of 2 indicates moderate con- 134 dition categories, a score of 3 signifies good condition categories and scores of 4 and 5 135 indicate obese condition categories. 136 All mule hooves were untrimmed.”

It should be described as detail as possible. What was the basis for these points? 

The statistic is not described. The sentence below is not enough: 

“One-way analysis of variance and a multivariate analysis 213 were used to explore the influence of sex, age and BCSs. The Tukey’s test was performed…”

Which traits were analyzed one-way? What was the model for multivariate? For what traits?

Author Response

Dear Reviewer 3

Thank you very much for your helpful information and suggestions. Now, we have addressed your suggestive and necessary points from your comment. I already attached the revised version with this box.

Thank you very much.

Kind regards,

Dr. Worakij Cherdchutham
